# Host-Dependent Variation in *Tetranychus urticae* Fitness and Microbiota Composition Across Strawberry Cultivars

**DOI:** 10.3390/insects16080767

**Published:** 2025-07-25

**Authors:** Xu Zhang, Hongjun Yang, Zhiming Yan, Yuanhua Wang, Quanzhi Wang, Shimei Huo, Zhan Chen, Jialong Cheng, Kun Yang

**Affiliations:** 1College of Agronomy and Horticulture, Jiangsu Vocational College of Agriculture and Forestry, Zhenjiang 212400, China; zhangxu@jsafc.edu.cn (X.Z.); hjyang@jsafc.edu.cn (H.Y.); yanzhim@jsafc.edu.cn (Z.Y.); wangyuanhua@jsafc.edu.cn (Y.W.); wangquanzhi@jsafc.edu.cn (Q.W.); 2015202030@njau.edu.cn (S.H.); jl1774079647@163.com (J.C.); 2Jiangsu Engineering and Technology Center for Modern Horticulture, Zhenjiang 212400, China; 3Shandong Province Centre for Bioinvasions and Eco-Security, Qingdao Agricultural University, Qingdao 266109, China; 18254995773@163.com; 4Shandong Engineering Research Center for Environment-Friendly Agricultural Pest Management, College of Plant Health and Medicine, Qingdao Agricultural University, Qingdao 266109, China

**Keywords:** *Tetranychus urticae*, strawberry cultivars, fitness, bacterial microbiota, *Wolbachia*

## Abstract

This study investigates how different strawberry cultivars affect the fitness and bacterial microbiota of the two-spotted spider mite, *Tetranychus urticae*. The research found that mites reared on the ‘Xuetu’ cultivar exhibited the highest fecundity, while those on ‘Xuelixiang’ had the shortest pre-adult development time. Additionally, ‘Xuelixiang’, ‘Xuetu’, and ‘Ningyu’ cultivars supported longer mite longevity compared to others. Microbiota analysis revealed that *Wolbachia* was the predominant bacterium across all mite groups, with varying abundances of other bacteria like *Pedobacter* and *Virgibacillus* depending on the strawberry cultivar. Notably, male mites exhibited higher bacterial diversity than females. These findings suggest that strawberry cultivar selection can influence mite fitness and microbiota composition, providing insights for developing integrated pest management strategies.

## 1. Introduction

*Tetranychus urticae Koch* (*Acari*: *Tetranychidae*), commonly known as the two-spotted spider mite, is a highly adaptable and polyphagous arthropod in the family Tetranychidae. This mite is capable of feeding on over 1200 plant species across more than 140 different plant families, including numerous crops and ornamental species [1,2]. It is considered one of the most economically significant pests due to its capacity to cause severe damage to plants through direct feeding [3]. Its rapid reproduction rate and ability to quickly develop resistance to pesticides complicate control efforts [4,5]. The mite’s ability to survive in diverse environments contributes to its global distribution.

Bacterial microbiota in arthropods refers to the diverse communities of bacteria residing within or on insect hosts, engaging in symbiotic relationships that range from mutualistic to parasitic. These microbiota are integral to various physiological, ecological, and evolutionary processes, influencing arthropod health, behavior, and adaptation [6,7,8,9]. Microbiota are vital for the growth, resistance, reproductive manipulations, and host adaptation of arthropods. For instance, in aphids, the primary symbiont *Buchnera* (an obligate endosymbiont) synthesizes essential amino acids missing in phloem sap [10], while *Wolbachia*-infected mosquitoes show elevated basal immunity and dramatically reduced dengue virus replication [7]. In *Drosophila*, secondary symbiont *Spiroplasma* protects against parasitoids, and *Acetobacter* affects behavior [11]; in beetles, *Nardonella* aids detoxification, and *Burkholderia* confers chemical resistance [12,13]. In *Bemisia tabaci* (Gennadius, 1889), secondary symbiont *Cardinium* infection would significantly increase the thermotolerance of host whiteflies [8]. All these studies highlight the vital effects of microbiota, especially symbiont infection for host arthropods.

In spider mites, the bacterial microbiota is abundant and complex, with significant functions. A study based on five *Tetranychus* species found that some bacterial symbionts showed host-species specificity [9]. In *Turtranychus Truncatus Ehara*, mites coinfected with secondary symbionts *Wolbachia* and *Spiroplasma* had significantly higher daily fecundity and juvenile survival rates than those infected with a single bacterium or uninfected [14]. *Wolbachia* also induces cytoplasmic incompatibility in *T. truncatus* by significantly down-regulating chorion proteins S38-like and Rop [15]. Infected with a single bacterium mites or uninfected mites exhibited reduced thermotolerance [16]. In *T. urticae*, salivary protein SHOT1s are positively correlated with *Wolbachia* density and account for *Wolbachia*-mediated phenotypes [17].

Plants have developed various defense mechanisms against herbivorous arthropods, including spider mites, such as physical barriers and chemical responses [18]. For example, plants can raise many Si-mediated physical, biochemical, and molecular mechanisms in defense against pathogens and insect pests [19]. Plants can synthesize toxic metabolites (alkaloids, glucosinolates, cyanogenic glycosides, phenolics) and anti-digestive proteins (protease inhibitors) that impair insect growth, survival, or reproduction [20,21]. Additionally, there is an evolutionary “war” between pests and host plants; for instance, *B. tabaci* acquired a horizontally transferred gene (*BtPMaT1*) from host tomato to neutralize plant toxins [22].

*Tetranychus urticae* is a major pest of strawberries, yet the influence of different strawberry cultivars on its fitness and microbiota remains poorly understood. While host plant traits are known to shape herbivore performance and microbial communities in other arthropod systems [6,14,16], cultivar-specific effects on *T. urticae* biology and microbiome dynamics are underexplored. This study addresses this gap by systematically evaluating how strawberry varietal differences modulate mite fitness and associated bacterial symbionts, with implications for integrated pest management strategies.

## 2. Materials and Methods

### 2.1. Collection and Cultivation of Tetranychus urticae

The *T. urticae* population utilized in this study was collected from Jurong, Jiangsu Province, China, in November 2024. Upon arrival at the laboratory, isofemale lines were established from individual females to ensure clonal purity. Given the haplodiploid reproductive system of *T. urticae*, all individuals within a line are progeny of a single mother. The mites were initially reared on leaves of the ‘Hongyan’ (HY) strawberry cultivar, sourced from a local plantation in Jurong City.

Mite colonies were maintained in 90 mm Petri dishes under controlled laboratory conditions: 25 ± 1 °C, 60% relative humidity, and a 16:8 light–dark photoperiod. Fresh strawberry leaves were carefully washed, sterilized at high temperature (55 °C, 30 min), and placed on circular sponges within the dishes. The leaves were oriented with their abaxial side facing upward. Water was added to fully saturate the sponges, ensuring the water level in the dishes exceeded the sponge height. Filter paper was trimmed to cover the leaf edges. After preparation, mite-infested leaf sections were observed under a microscope, and areas with high mite density were cut and transferred for colony expansion. Labels were attached to track each colony. Water was replenished every 1–2 days, and dishes were replaced every 4–5 days.

### 2.2. Fitness Assessment of T. urticae on Different Strawberry Cultivars

#### 2.2.1. Pre-Adult Development Period and Longevity

To assess the effects of various strawberry cultivars on the fitness of *T. urticae*, seven cultivars were selected: Hongyan (HY), Yuexiu (YX), Tianshi (TS), Ningyu (NY), Xuetu (XT), Zhangjj (ZJ), and Xuelixiang (XLX). All strawberry cultivars were planted in laboratory at 25 ± 1 °C, 60% relative humidity, and a 16:8 light–dark photoperiod. For fitness assessments, mites were transferred to each of the seven test cultivars (HY, YX, TS, NY, XT, ZJ, XLX) and reared for two complete generations to record the fecundity, pre-adult period and longevity of spider mites. The fecundity, pre-adult period and longevity-tested mites were F2 offspring reared exclusively on their respective test cultivar. This ensured physiological adaptation to cultivar-specific traits before measurements. Experiments were conducted in an incubator maintained at 25 °C and 80% relative humidity. Leaves from the same nodal position of each cultivar were selected, inspected under a microscope, and cleaned of pests and predators using a soft brush. Each Petri dish was divided into four sectors using paper strips. One female mite and 2–3 male mites were placed in each sector. After oviposition, adults were removed, and egg hatching was monitored. Post-hatching, daily observations recorded mite lifespan, encompassing both nymphal and adult stages.

#### 2.2.2. Fecundity Assessment

To evaluate the fecundity of *T. urticae* on different strawberry cultivars, experiments were performed under identical environmental conditions (25 °C, 80% relative humidity). Leaves were prepared as described above. Each Petri dish was divided into four sectors, with six replicate dishes per treatment group. One quiescent deutonymph-stage female and 2–3 male mites were introduced into each sector. Daily egg production was recorded starting from the first oviposition event.

### 2.3. Microbiota Analysis of T. urticae on Different Strawberry Cultivars

The bacterial microbiota of 1-day-old female and male *T. urticae* adults reared on four strawberry cultivars (TS, HY, FY, and YX) were analyzed using 16S rRNA gene sequencing, as the four cultivars were most widely grown commercial varieties in China, especially in Jiangsu Province. Each sample including 20 female/male spider mite adults. Genomic DNA was extracted from samples using the OMEGA Soil DNA Kit (M5635-02) (Omega Bio-Tek, Norcross, GA, USA), adhering to the manufacturer’s protocol. DNA concentration and purity were assessed using a NanoDrop NC2000 spectrophotometer (Thermo Scientific, Waltham, MA, USA), with A260/A280 ratios confirming protein-free extracts, while integrity was verified via 1% agarose gel electrophoresis.

For bacterial 16S rRNA gene sequencing, nearly full-length amplicons (~1500 bp) were generated using primers 27F (5′-AGAGTTTGATCMTGGCTCAG-3′) and 1492R (5′-ACCTTGTTACGACTT-3′) in a two-step PCR process. The first PCR step included 25 cycles to amplify target regions, followed by a second 10-cycle PCR to incorporate 16 bp sample-specific barcodes. Reaction mixtures contained 5 μL Q5 reaction buffer (5×), 5 μL Q5 High-Fidelity GC buffer (5×), 0.25 μL Q5 DNA Polymerase (5 U/μL), 2 μL dNTPs (2.5 mM), 1 μL each primer (10 μM), 2 μL DNA template, and 8.75 μL ddH_2_O. Thermal cycling conditions were: 98 °C for 2 min (initial denaturation), followed by 25/10 cycles (first/second PCR) of 98 °C for 30 s, 55 °C for 30 s (annealing), and 72 °C for 90 s (extension), with a final 5 min extension at 72 °C. Amplicons were purified using Agencourt AMPure Beads (Beckman Coulter, Lane Cove West, NSW, Australia), quantified via PicoGreen dsDNA Assay, and pooled equimolarly for Single Molecule Real-Time (SMRT) sequencing on the PacBio Sequel platform. Circular consensus sequencing (CCS) was employed to minimize errors, generating high-accuracy reads (≥3 passes, ≥99% predicted accuracy). Raw sequences were processed using SMRT Link (v5.0.1.9585), trimmed to ≤2000 bp, and filtered to exclude low-quality reads.

Bioinformatic analysis was conducted in QIIME2 (2022.11). Demultiplexed reads were primer-trimmed with Cutadapt, denoised via DADA2, and aligned using MAFFT to generate amplicon sequence variants (ASVs). A phylogenetic tree was constructed with FastTree2, and taxonomy was assigned against the SILVA 138 (bacteria)databases. Alpha diversity metrics (Chao1, Shannon, Simpson, Faith’s PD) and beta diversity (Bray–Curtis, weighted/unweighted UniFrac) were calculated after rarefaction. Community structure differences were visualized via PCoA, NMDS, and UPGMA clustering, with statistical validation using PERMANOVA, ANOSIM, and LEfSe. Machine learning approaches, including Random Forest (5-fold cross-validation) and OPLS-DA, identified discriminative taxa. Co-occurrence networks were built using SparCC (pseudocount = 10^−6^, correlation cutoff = 0.7) and visualized with igraph. Functional profiling leveraged PICRUSt2 to predict metabolic pathways against MetaCyc and KEGG databases. All analyses were performed in R (v3.2.0) with packages like phyloseq and vegan, ensuring reproducibility and statistical rigor. While the data of fecundity and pre-adult period of *T. urticae* reared on different strawberry cultivars were first tested for normality (Kolmogorov—Smirnov test) and homogeneity of group variances (Levene’s test). As the fecundity data follows a normal distribution, they were analyzed with a one-way ANOVA with posthoc Tukey HSD analysis. Pre-adult period data do not follow a normal distribution; the data were ‘transformed withy=log10(x)
to make sure the data follow a normal distribution, and then the data of pre-adult period were analyzed with a one-way ANOVA with posthoc Tukey HSD analysis. The longevitiy (survival assay) of *T. urticae* reared on different strawberry cultivars was compared by the Kaplan–Meier method. SPSS 19 (IBM, Armonk, NY, USA) was used to carry out all statistical tests.

## 3. Results

### 3.1. Tetranychus urticae Fitness on Various Strawberry Cultivars

The fitness parameters of *T. urticae* were assessed on seven strawberry cultivars: Hongyan (HY), Yuexiu (YX), Tianshi (TS), Ningyu (NY), Xuetu (XT), Zhangjj (ZJ), and Xuelixiang (XLX). Among these, XT and NY cultivars supported the highest fecundity, with an average of 166.56 eggs per female, significantly surpassing all other cultivars (One-Way ANOVA, *p* < 0.05) (Figure 1).

The pre-adult periods of *T. urticae* reared on different cultivars were compared (Figure 2). Based on the results, pre-adult period of *T. urticae* reared on XLX breed (7.71 ± 0.13 days, Mean ± SEM) was significantly shorter than those reared on TS (10.11 ± 0.24, Mean ± SEM), XT cultivars (9.25 ± 0.37 days) and NY cutivars (9.00 ± 0.42 days) (One-Way ANOVA, *p* < 0.05).

Based on Log-rank (Mantel–Cox) test, for longevity of *T. urticae* reared on different cultivars, results showed that longevity of *T. urticae* reared on strawberry cultivars XLX (25.95 ± 1.39, Mean ± SEM), XT (26.83 ± 0.87) and NY (26.83 ± 1.03) were significantly longer than that of *T. urticae* reared on other strawberry cultivars (Figure 3).

### 3.2. Tetranychus urticae Bacterial Microbiota Changed in Various Strawberry Cultivars

#### 3.2.1. Sequencing Data of Bacterial Microbiota in *Tetranychus urticae*

The bacterial microbiota of 1-day-old female and male *T. urticae* adults reared on 4 strawberry breeds (including TS, HY, FY and YX) were measured by 16s rRNA gene sequencing. Based on results, in different *T. urticae* samples reared on various strawberry cultivars, the ASVs (amplicon sequence variants) numbers ranged from 9454 to 15,975, and the denoised ASVs numbers ranged from 9369 to 15,942, all *T. urticae* samples including total 9 bacterial phyla, all samples ranged from 2 to 6 bacterial phyla, 4 to 38 bacterial species (Table 1). Based on Principal Coordinates Analysis (PCoA) all samples from different growth stages clustered together, which showed insignificant influence of different *T. urticae* groups reared on various strawberry cultivars on structures of microbiota, except for YXF group (Appendix A). For the top 19 bacterial genera of *T. urticae*, most bacterial genera were related to phyla Proteobacteria, Firmicutes and Actinobacteria (Figure 4).

#### 3.2.2. Bacterial Communities in *Tetranychus urticae* Reared on Different Strawberry Cultivars

Based on 16s rRNA gene sequencing, in *T. urticae*, Proteobacteria bacteria occupy the largest proportion (over 89.96% in all groups) of the total bacteria abundance across all *T. urticae* groups reared on different strawberry cultivars, while Bacteroidetes bacteria are the second most abundant (Appendix A). Regarding bacterial species, the secondary symbiont, *Wolbachia symbiont of T. urticae,* is the most dominate bacterium, occupying more than 89.58% of the bacterial abundance in all *T. urticae* groups. *Pedobacter sp. An13* and *Virgibacillus halodenitrificans* are abundant in some *T. urticae* groups reared on different strawberry cultivars (Figure 5). For *Wolbachia*, although the proportions varied in *T. urticae* groups reared on different strawberry cultivars ranged from YXF group (89.58 ± 3.82%, Mean ± SEM) to TSF (99.19 ± 0.28%), there were no significant differences among all *T. urticae* groups (Kruskal–Wallis test, *p* = 0.074) (Appendix A), the *Wolbachia* proportion of male adults (94.02 ± 1.31%) was not significantly different than that of female adults (96.55 ± 1.38%) (Student’s *t*-test, *t* = −1.331, df = 30, *p* = 0.19) (Appendix A).

#### 3.2.3. Bacterial Microbiota Varied in *Tetranychus urticae* Reared on Different Strawberry Cultivars

The bacterial diversity across *T. urticae* reared on different strawberry cultivars was assessed using 16S rRNA gene sequencing. In total, the diversity index of male adults were significantly higher than that of female adults (Appendix A). Shannon index of male adults were significantly higher than that of female adults (Student’s *t*-test, *t* = 2.655, df = 30, *p* < 0.05), similar results detected in Chao1 index, as Chao1 index of *T. urticae* male adults were significantly higher than that of female adults (Student’s *t*-test, *t* = 3.376, df = 30, *p* < 0.01), while the Simpson index between *T. urticae* male and female adults were not significantly different (Student’s *t*-test, *t* = 3.376, df = 30, *p* = 0.12) (Appendix A).

As for diversity index of *T. urticae* on different strawberry cultivars, for Shannon indexes, the analysis revealed that the HYM group (0.69 ± 0.23, Mean ± SEM) and YXM group (0.69 ± 0.25) were significantly higher than that of TSF, HYF, FYM and FYF groups (Figure 6A). For Simpson indexes, the analysis revealed that the YXF group (0.18 ± 0.06), HYM group (0.16 ± 0.06) and YXM group (0.15 ± 0.06) were significantly higher than that of TSF and FYF groups (Figure 6B). For Chao1 indexes, the analysis revealed that the TSM group (30.13 ± 9.88), HYM group (30.77 ± 3.76) and YXM group (30.12 ± 8.78) were significantly higher than that of TSF and FYF groups (One-Way ANOVA, *p* < 0.05) (Figure 6C).

Based on bacterial microbiota function prediction results, most bacterial genes were related to Biosynthesis, while other bacterial genes were related to Degradation/Utilization/Assimilation, Detoxification, Generation of Precursor Metabolites and Energy, Glycan Pathways, Macromolecule Modification and Metabolic Clusters (Figure 7).

## 4. Discussion

The two-spotted spider mite, *T. urticae*, is a formidable pest with rapid reproductive rates and adaptive plasticity, enabling it to exploit diverse host plants. This study reveals critical impacts of strawberry cultivar variation on *T. urticae* biology. We demonstrate that cultivar traits significantly modulate (i) mite fitness parameters and (ii) associated bacterial microbiota composition. These findings provide actionable insights for developing integrated pest management (IPM) strategies.

### 4.1. Influence of Strawberry Cultivars on T. urticae Fitness

The observed variations in fecundity, development time, and longevity of *T. urticae* across different strawberry cultivars highlight the role of host plant characteristics in shaping pest biology. Notably, mites reared on the XT variety exhibited the highest fecundity, while those on the XLX variety had the shortest pre-adult period. These differences may be attributed to the varying nutritional profiles and secondary metabolite compositions of the cultivars, which can affect mite development and reproduction [23,24,25,26]. In western flower thrips *Frankliniella occidentalis* Pergande reared on different *Rosa chinensis* Jacq. cultivars, which F. occidentalis showed digestive enzyme activities in various cultivars [26]. Our findings also aligned with previous research indicating that plant genotype can influence herbivore performance through mechanisms like antibiosis, where plant compounds adversely affect pest biology [27].

### 4.2. Microbiota Composition and Its Modulation by Host Plants

The dominance of Proteobacteria, particularly *Wolbachia*, across all mite groups suggests a stable core microbiota in *T. urticae*, which results were consistent to other related research [28,29,30]. However, the presence of other bacterial taxa, such as Pedobacter and *Virgibacillus*, varied among mites reared on different cultivars, indicating that host plant species can modulate the peripheral microbiota, microbiota diversities also varied in *T. urticae* reared on different strawberry cultivars. This modulation may result from differences in plant chemistry, which can influence microbial acquisition and retention [31,32,33]. For example, host plants not only affected the biological characteristics and nutritional metabolism of pea aphids but also regulated the symbiotic density [34]. A secondary metabolite from cotton, Gossypol, suppressed *Buchnera* populations in cotton-melon aphid, *Aphis gossypii* [35]. The difference in plant chemistry among different strawberry cultivars may cause the changes in microbiota in *T. urticae*, which should be further explored. The role of *Wolbachia* in enhancing host detoxification pathways and conferring pesticide resistance has been documented [30]. *Wolbachia* thus serves as a foundational detoxifier, while cultivar-driven changes in ancillary bacteria may augment or impair this function. Resistant cultivars could exploit this by simultaneously challenging *Wolbachia* and disrupting supportive microbiota.

Functional profiling via PICRUSt2 predicted potential roles of the mite microbiota, primarily in biosynthesis, with secondary involvement in detoxification and energy metabolism (Figure 7). However, these inferences carry inherent limitations: Recent evaluations confirm that 16S-based functional tools (including PICRUSt2) lack sensitivity to detect biologically meaningful metabolic shifts due to incomplete reference databases and inability to resolve strain-level functions [36]. Thus, while these predictions hypothesize microbiota contributions to mite adaptation (e.g., detoxifying cultivar-specific plant compounds), they require validation. We advocate for metatranscriptomic/metabolomic approaches in future work to directly quantify microbial functions in *T. urticae*.

### 4.3. Sex-Based Differences in Microbial Diversity

The higher bacterial diversity observed in male mites compared to females raises questions about sex-specific interactions between *T. urticae* and its microbiota. These differences could stem from variations in behavior, physiology, or immune responses between sexes, which in turn affect microbial colonization and maintenance [37,38,39,40]. Understanding microbiota dynamics is crucial because cultivar-driven shifts correlate with fitness outcomes (Figure 1, Figure 2, Figure 3, Figure 5 and Figure 6), and predicted functions (e.g., detoxification; Figure 7) align with known symbiont roles in T. urticae [28,29,30]. While future studies should quantify enzymatic/metabolic consequences, our work establishes host plant variety as a key modulator of mite-microbiota functional interactions.

### 4.4. Implications for Integrated Pest Management

The study’s findings have practical implications for developing sustainable pest management strategies. By identifying strawberry cultivars that significantly reduce *T. urticae* fitness (e.g., ‘Tianshi’ prolonging development by 31% and ‘Zhangji’ reducing fecundity vs. ‘Xuetu’), breeders can prioritize these varieties to mitigate pest damage. Furthermore, host plants concurrently altered mite microbiota composition (Figure 5 and Figure 6), suggesting that cultivars favoring disadvantageous microbial shifts could synergistically enhance resistance—though causal links require experimental validation.

Separately, targeting conserved symbionts like *Wolbachia* (89–99% abundance) may offer control avenues, given its established roles in pesticide detoxification [30] and reproduction [17] in spider mites. By targeting *Wolbachia*, researchers could offer novel avenues for controlling mite populations [40,41,42]. Such approaches align with IPM principles, combining host resistance, biological tools, and targeted interventions.

### 4.5. Future Research Directions

Further studies are needed to elucidate the mechanisms underlying the interactions between host plants, *T. urticae*, and its microbiota. Investigating the functional roles of specific microbial taxa and their contributions to mite physiology and adaptability will enhance our understanding of these complex relationships. Additionally, exploring the potential for microbiota manipulation as a pest control strategy warrants attention, particularly in the context of developing resistance management programs.

In conclusion, this study tested and confirmed that strawberry cultivar traits significantly alter *T. urticae* fitness, and further demonstrate that these fitness differences correlate with host plant-induced microbiota shifts, evidenced by: (i) Microbiota diversity reductions in mites from poor-performance cultivars (Figure 6). (ii) Conserved *Wolbachia* dominance alongside cultivar-sensitive taxa (Figure 5). (iii) Predicted microbial functions mirroring fitness trends. These findings objectively inform breeding programs and reveal microbiota as a testable target for integrated pest management.

## Figures and Tables

**Figure 1 insects-16-00767-f001:**
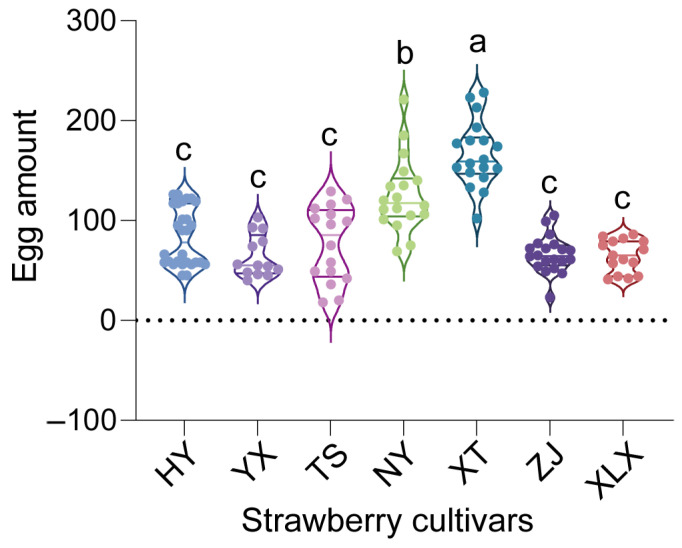
Fecundity of *Tetranychus urticae* on seven Chinese strawberry cultivars: Hongyan (HY), Yuexiu (YX), Tianshi (TS), Ningyu (NY), Xuetu (XT), Zhangjj (ZJ), and Xuelixiang (XLX). Different letters indicate significant differences as determined by One-Way ANOVA (*p* < 0.05).

**Figure 2 insects-16-00767-f002:**
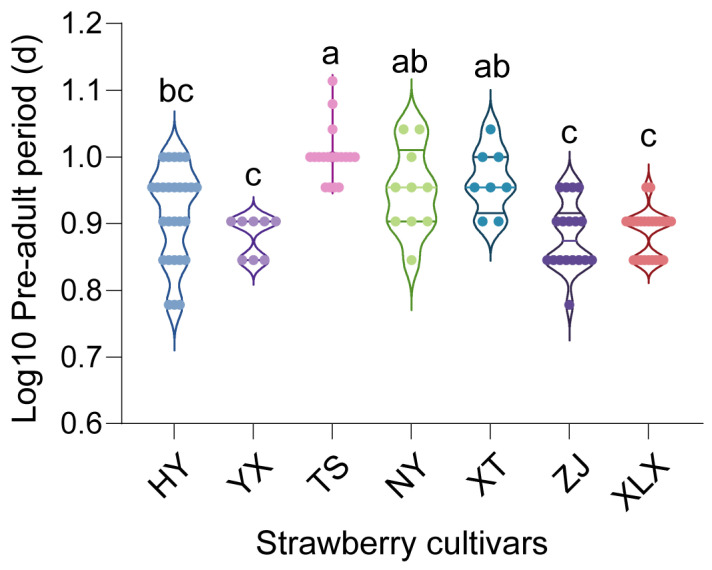
Log10 transformation of pre-adult periods of *Tetranychus urticae* on seven Chinese strawberry cultivars: Hongyan (HY), Yuexiu (YX), Tianshi (TS), Ningyu (NY), Xuetu (XT), Zhangjj (ZJ), and Xuelixiang (XLX). Different letters indicate significant differences as determined by One-Way ANOVA (*p* < 0.05).

**Figure 3 insects-16-00767-f003:**
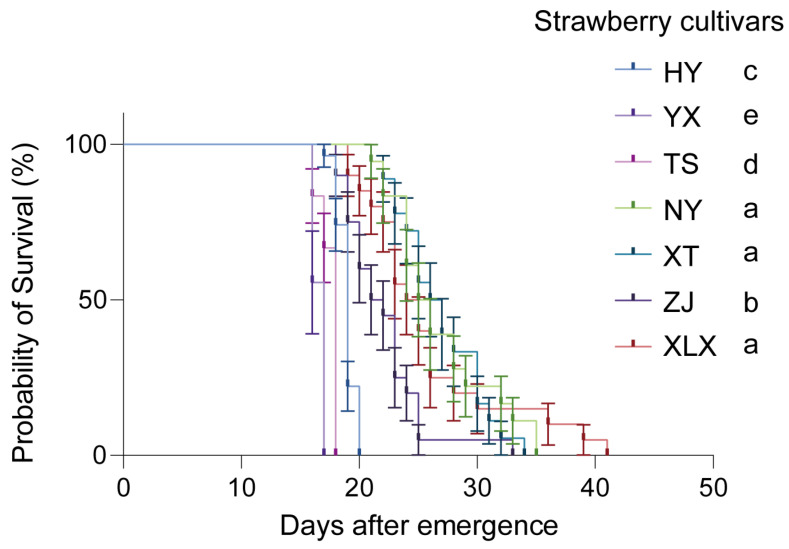
Survivorship curve of *Tetranychus urticae* on seven Chinese strawberry cultivars: Hongyan (HY), Yuexiu (YX), Tianshi (TS), Ningyu (NY), Xuetu (XT), Zhangjj (ZJ), and Xuelixiang (XLX). Different letters indicate significant differences as determined by Log-rank (Mantel–Cox) test (*p* < 0.05).

**Figure 4 insects-16-00767-f004:**
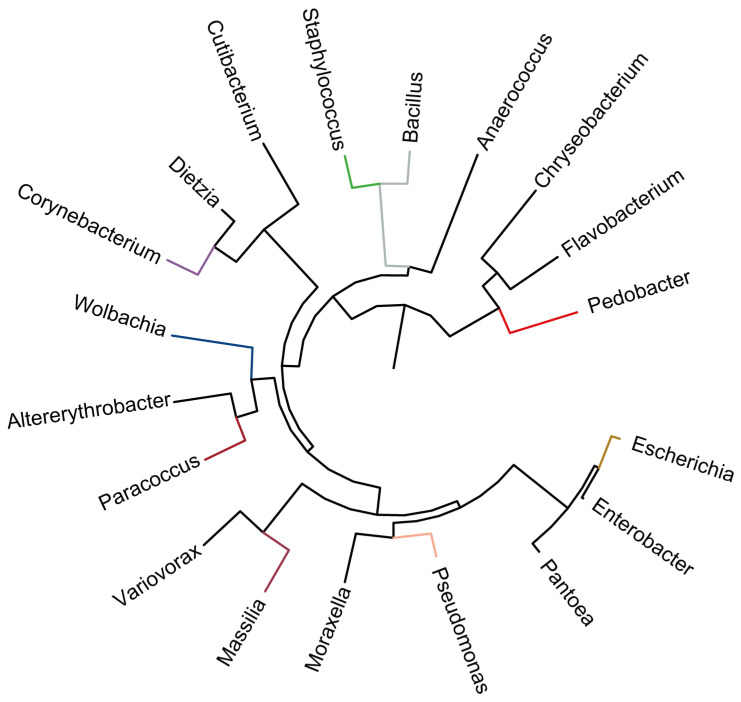
Phylogenetic tree of top 19 abundant bacteria in *Tetranychus urticae* 1-day-old male/female adults on 4 Chinese strawberry cultivars Tianshi (TS), Hongyan (HY), Ningyu (NY), Fenyu (FY) and Yuexiu (YX), by 16S rRNA genes based on probabilistic methods of phylogenetic inference constructed by FastTree 2.0.0 software.

**Figure 5 insects-16-00767-f005:**
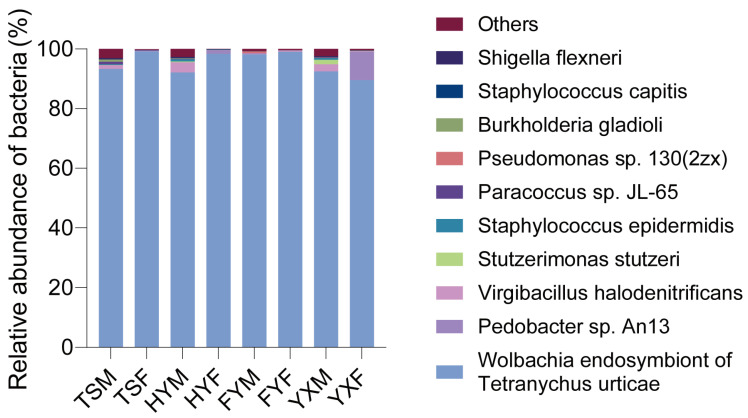
Relative abundance of top 10 bacterial species in *Tetranychus urticae* 1-day-old male/female adults on 4 Chinese strawberry cultivars Tianshi (TS), Hongyan (HY), Ningyu (NY), Fenyu (FY) and Yuexiu (YX), by full-length 16s rRNA gene sequencing. In the group labels, the third letter denotes the sex of the mites: ‘F’ for female and ‘M’ for male.

**Figure 6 insects-16-00767-f006:**
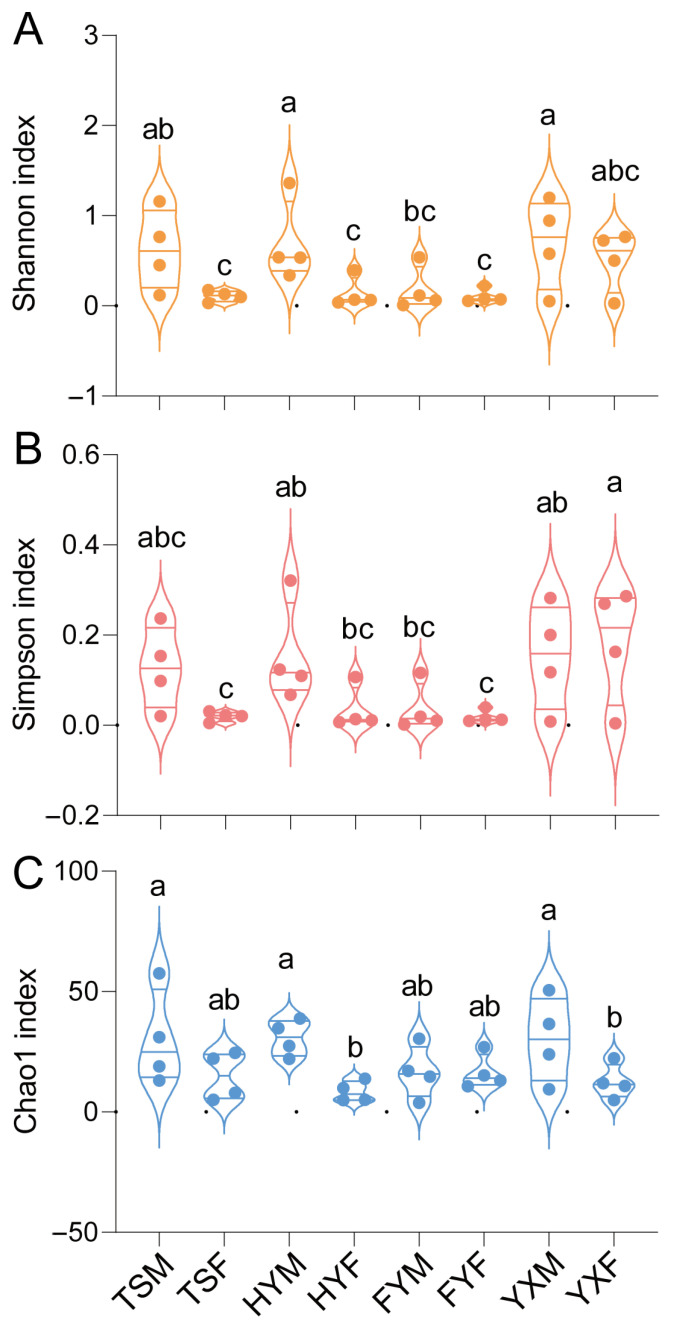
Alpha diversity indices of bacterial communities in 1-day-old male and female *Tetranychus urticae* adults reared on four Chinese strawberry cultivars: Tianshi (TS), Hongyan (HY), Fenyu (FY), and Yuexiu (YX). The diversity metrics presented include Shannon (**A**), Simpson (**B**), and Chao1 (**C**) indices. In the group labels, the third letter denotes the sex of the mites: ‘F’ for female and ‘M’ for male. Different lowercase letters above the bars (a, b, c) indicate statistically significant differences between groups.

**Figure 7 insects-16-00767-f007:**
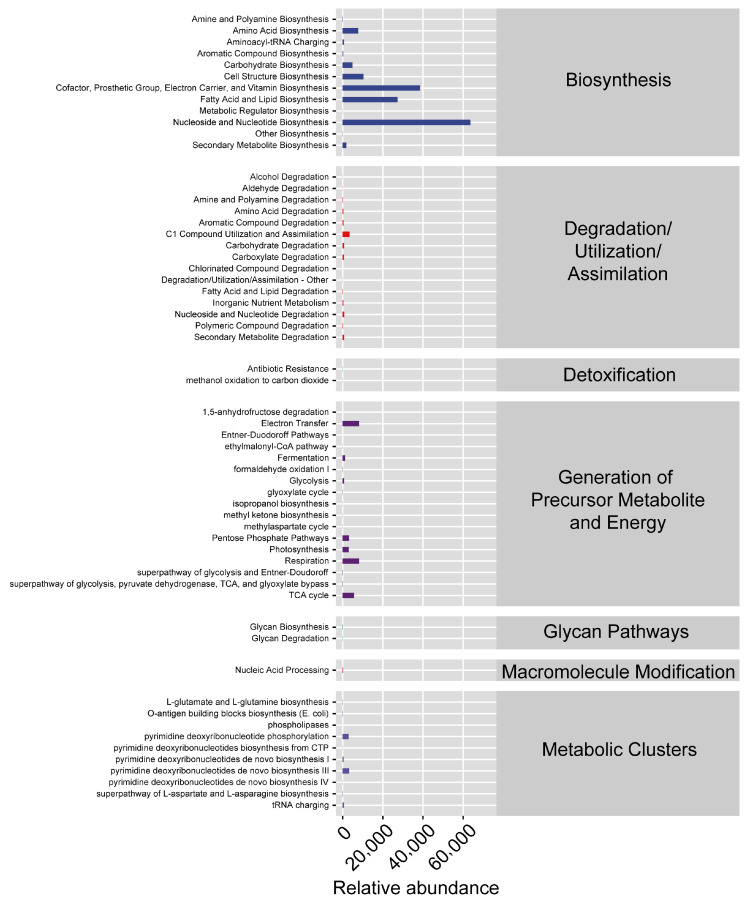
Bacterial microbiota function prediction of *Tetranychus urticae* reared on four Chinese strawberry cultivars: Tianshi (TS), Hongyan (HY), Fenyu (FY), and Yuexiu (YX), by 16s rRNA gene sequencing. Functional profiling leveraged PICRUSt2 to predict metabolic pathways against MetaCyc and KEGG databases.

**Table 1 insects-16-00767-t001:** Information about full-length 16s rRNA gene sequencing in different strawberry cultivars of *Tetranychus urticae*.

Sample Name	Input	Filtered	Denoised	Phylum	Species
TSM1	9454	9454	9369	4	9
TSM2	10,303	10,303	10,196	4	21
TSM3	14,999	14,999	14,876	4	38
TSM4	15,463	15,463	15,387	5	16
TSF1	15,050	15,050	14,992	5	20
TSF2	14,562	14,562	14,555	2	4
TSF3	14,019	14,019	13,936	6	20
TSF4	13,536	13,536	13,524	4	7
HYM1	14,520	14,520	14,486	3	14
HYM2	14,892	14,892	14,512	4	25
HYM3	13,410	13,410	13,348	4	17
HYM4	15,636	15,636	15,502	5	25
HYF1	12,979	12,979	12,965	4	10
HYF2	12,687	12,687	12,663	4	9
HYF3	12,352	12,352	12,282	2	4
HYF4	14,573	14,573	14,570	2	4
FYM1	13,989	13,989	13,952	2	4
FYM2	15,759	15,759	15,593	4	11
FYM3	15,138	15,138	15,080	4	21
FYM4	14,878	14,878	14,793	5	13
FYF1	14,918	14,918	14,862	4	11
FYF2	12,277	12,277	12,194	4	22
FYF3	13,397	13,397	13,349	4	9
FYF4	13,533	13,533	13,457	4	11
YXM1	15,624	15,624	15,614	4	9
YXM2	14,113	14,113	14,102	3	11
YXM3	15,449	15,449	15,061	5	32
YXM4	14,191	14,191	14,144	5	24
YXF1	15,636	15,636	15,610	2	4
YXF2	14,354	14,354	14,291	4	15
YXF3	13,540	13,540	13,477	4	11
YXF4	15,975	15,975	15,942	4	11

## Data Availability

Sequence data that support the findings of this study have been deposited in the NCBI with the primary accession code SUB15348674.

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
