# Peer review of "Host-Dependent Variation in Tetranychus urticae Fitness and Microbiota Composition Across Strawberry Cultivars"

_insects, 2025, doi:10.3390/insects16080767_

Round 1

Reviewer 1 Report

Comments and Suggestions for Authors

Tetranychus urticae is a global pest, highly polyphagous and hazardous to numerous crops. It’s control with synthetic insecticides, just like in other pests, unleashes serious problems concerning safety (due to environmental pollution), health risk (due to food contamination), and efficiency (due to fast developing resistance). Numerous research groups dedicate efforts to elucidating mite biology, interactions with host plants, and control strategies. Thus, the paper under review will drive attention of the journal’s audience. It can be considered for publication, but requires to be made clearer and more straightforward, with each part fitting the others.

I tried to list the main issues below for Major Revision.

In Introduction, more examples could be given from different arthropod species about importance of microbiome and particular endosymbiotic bacteria

In Methods, it should be clearly explained for how long the cultivation of mites on different cultivars was performed, in terms of both time and number of generations.

L68 and elsewhere: singly infected = infected with a single bacterium

L69-70: what does it mean “to regulate proteins”?

L85-93: no need to disclose experimental design and key findings in the Introduction

L96 and further: after first mentioning of the genus epithet in full in the main text, it should be contracted throughout

L107: it is not clear how the filter paper is used and does it touch the water? Can you add a picture?

L119: how do you ensure absence of contacts between mites, separated by paper strips? Can you add a picture?

L173-175: why means are given without SEM here?

L182: what is the measurement unit for pre-adult period?

L183: after first introduction of “Mean ± SEM”, it can be omitted in further value sets

L206-207: all samples … all clustered – sounds as a tautology

Figure 2: What is the reason for performing phylogenetic reconstruction in the frames of the current study?

L217 and possibly elsewhere: check italics for Latin genus and species epithets throughout

L243: the microbiota … was higher – what do you mean?

L250-256: Shannon Index might have been higher, but not the group itself.

L264-273: what is the reason of this part of work? How does it relate to the study of strawberry cultivar effects on mite fitness and microbial community?

Fig. 7: where the strawberry cultivars are shown in the picture?

Ibidem: which units the Relative Abundance is measured in?

L308-310: but what is the relation between microbiota modulation by strawberry cultivars and Wolbachia role in detoxification?

L335: please elaborate which beneficial microbial interactions are presumed to be disrupted in the present study?

L336-338: how targeting Wolbachia relates to the current study?

Comments on the Quality of English Language

L192: longer than T. urticae = longer than that of T. urticae

L201: bredds = breeds

L205: 9 phylum bacteria = 9 bacterial phyla

L209: genus = genera

L221: bacteria is= bacteria are

L223: bacteria = bacterium

L228: were no significant difference = were no significant differences

L278-280: the sentence is rather long. Consider breaking down

L297: researches = research

Author Response

  1. In Introduction, more examples could be given from different arthropod species about importance of microbiome and particular endosymbiotic bacteria

A:  We sincerely appreciate this constructive suggestion. To better contextualize the role of microbiomes in arthropods, we have expanded the Introduction with additional examples that highlight the functional diversity of endosymbiotic bacteria across taxa. In Introduction we added a more examples in 2nd paragraph to describe the importance of bacterial microbiota in arthropods.

  1. In Methods, it should be clearly explained for how long the cultivation of mites on different cultivars was performed, in terms of both time and number of generations.

A:  Detailed information added in section 2.2.1.

  1. L68 and elsewhere: singly infected = infected with a single bacterium

A:  Corrected.

  1. L69-70: what does it mean “to regulate proteins”?

A:  Chorion protein S38-like and Rop were down-regulated by Wolbachia, detailed information added in manuscript.

  1. L85-93: no need to disclose experimental design and key findings in the Introduction.

A:  This paragraph rewrite to focuse on broader context rather than specific results.

  1. L96 and further: after first mentioning of the genus epithet in full in the main text, it should be contracted throughout

A:  Corrected.

  1. L107: it is not clear how the filter paper is used and does it touch the water? Can you add a picture? L119: how do you ensure absence of contacts between mites, separated by paper strips? Can you add a picture?

A:  A:  The experiments did as the figure follows:

  1. L173-175: why means are given without SEM here?

A: Detailed information added in section 3.1.

  1. L182: what is the measurement unit for pre-adult period?

A:  The unit is “day”, and information added in manuscript.

  1. L183: after first introduction of “Mean ± SEM”, it can be omitted in further value sets

A:  Corrected.

  1. L206-207: all samples … all clustered – sounds as a tautology

A:  Delete the 2nd “all”. 

  1. Figure 2: What is the reason for performing phylogenetic reconstruction in the frames of the current study?

A:  The phylogenetic reconstruction in our study was performed to enhance our understanding of the bacterial diversity within Tetranychus urticae and its symbiotic microbiota. It helped confirm the taxonomic identity of key bacteria, such as Wolbachia, and allowed us to assess the evolutionary relationships among bacterial species. This approach also facilitated the examination of bacterial strain variability across different strawberry cultivars and mite sexes, providing valuable insights into the microbial diversity and structure within the mite population.

  1. L217 and possibly elsewhere: check italics for Latin genus and species epithets throughout

A:  Corrected throughout the manuscript.

  1. L243: the microbiota … was higher – what do you mean?

A: Thanks for your comments, and this should be “microbiota diversity”, contents corrected in manuscript.

  1. L250-256: Shannon Index might have been higher, but not the group itself.

A:  Related contents revised based on your comments.

  1. L264-273: what is the reason of this part of work? How does it relate to the study of strawberry cultivar effects on mite fitness and microbial community?

A:  We included alpha diversity analyses (Figure 6) and functional predictions (Figure 7) to demonstrate how strawberry cultivars holistically influence mite biology beyond fitness parameters. While Wolbachia dominance was consistent (Figure 5), Figure 6 reveals significant cultivar-driven variation in overall bacterial diversity, particularly higher male microbiota diversity on HY and YX cultivars, confirming host plants quantitatively reshape microbial communities. Figure 7 links these shifts to potential physiological functions: roles in detoxification and nutrient assimilation suggest microbiota may aid mites in overcoming cultivar-specific defenses or enhancing resource use, directly explaining observed fitness differences (such as the extended longevity on XLX/XT). Together, these analyses bridge microbiota composition and function to the study’s core objective, elucidating how host plants influence mite adaptability through both direct and microbially mediated pathways.

  1. 7: where the strawberry cultivars are shown in the picture?

A:  This is the summary of all bacterial microbiota function predictions of T. urticae reared on all strawberry cultivars. If necessary, we will provide the detailed information of function predict of different cultivars’ T. urticae.

  1. Ibidem: which units the Relative Abundance is measured in?

A:  Based on section 2.3, relative abundance is the proportion of the CCS counts of the bacteria in all bacterial communities.

  1. L308-310: but what is the relation between microbiota modulation by strawberry cultivars and Wolbachia role in detoxification?

A:  Detailed information added in section 4.2.

  1. L335: please elaborate which beneficial microbial interactions are presumed to be disrupted in the present study?

A:  This paragraph rewrite based on your and other reviewers’ comments.

  1. L336-338: how targeting Wolbachia relates to the current study?

A:  This paragraph rewrite based on your and other reviewers’ comments.

  1. L192: longer than T. urticae = longer than that of T. urticae

A:  Corrected.

  1. L201: bredds = breeds

A: Corrected.

  1. L205: 9 phylum bacteria = 9 bacterial phyla

A: Corrected.

  1. L209: genus = genera

A: Corrected.

  1. L221: bacteria is= bacteria are

A: Corrected.

  1. L223: bacteria = bacterium

A: Corrected.

  1. L228: were no significant difference = were no significant differences

A: Corrected.

  1. L278-280: the sentence is rather long. Consider breaking down

A: Corrected.

  1. L297: researches = research

A: Corrected.

Reviewer 2 Report

Comments and Suggestions for Authors

This manuscript presents a comprehensive study on the interaction between Tetranychus urticae (the two-spotted spider mite), and different strawberry cultivars, focusing on both fitness parameters and associated bacterial microbiota. The authors found that the performance of T. urticae varies significantly depending on the cultivar, affecting fecundity, development, and longevity. Microbiota composition, particularly the prevalence of Wolbachia, also varied subtly across host plants and sexes.

The paper follows logical progression, with clearly defined sections for introduction, methodology, results, and discussion. The methodology is sound, and the data are well-presented, supporting the research findings.

Some corrections should improve clarity and methodological transparency of the paper.

1) All scientific names should be italicized throughout the manucript and supplementary file.

2) The citation "Islam et al. 2020" should be added in the reference list.

3) Full scientific names of Tetranychus urticae, Tetranychus truncatus, Bemisia tabaci, Frankliniella occidentalis, Rosa chinensis should be provided the first time presented in the text.

4) References should be presented according to the format of Insects journal.

5) A clarification is necessary whether the mites used for the fecundity experiments were obtained from rearings on each strawberry cultivar of from the stock colony. 

6) A clarification is necessary about the number of females and males per cultivar were used for the mocrobiota analysis.

For further specific comments please see the attached files.

Author Response

Reviewer 2#

The paper follows logical progression, with clearly defined sections for introduction, methodology, results, and discussion. The methodology is sound, and the data are well-presented, supporting the research findings.

Some corrections should improve clarity and methodological transparency of the paper.

  • All scientific names should be italicized throughout the manucript and supplementary file.

A: Thanks for your comments, and we corrected all scientific names.

  • The citation "Islam et al. 2020" should be added in the reference list.

A: Cited reference added.

  • Full scientific names of Tetranychus urticaeTetranychus truncatusBemisia tabaciFrankliniella occidentalisRosa chinensisshould be provided the first time presented in the text.

A: Corrected.

  • References should be presented according to the format of Insects 

A: Thanks for your comments, all reference presented corrected.

  • A clarification is necessary whether the mites used for the fecundity experiments were obtained from rearings on each strawberry cultivar of from the stock colony. 

A: Detailed information added in section 2.2.1.

  • A clarification is necessary about the number of females and males per cultivar were used for the mocrobiota analysis.

A: Detailed information added in section 2.3.

  • For further specific comments please see the attached files.

A: All corrections added based on your comments in manuscript.

Reviewer 3 Report

Comments and Suggestions for Authors

This manuscript presents interesting and potentially valuable data. However, there are numerous issues that the authors need to address before the manuscript can be considered for further evaluation. Please refer to the attached PDF for detailed, section-specific comments and suggestions.

Author Response

Reviewer 3#

This manuscript presents interesting and potentially valuable data. However, there are numerous issues that the authors need to address before the manuscript can be considered for further evaluation. Please refer to the attached PDF for detailed, section-specific comments and suggestions.

A: Thanks for your comments, all corrections added based on your comments in manuscript.

  1. All scientific names added in manuscript.
  2. All strawberry cultivars planted in lab, detailed information added in section 2.2.1.
  3. NY, YX, TS and HY were selected for 16s rRNA gene sequencing due to their economic importance in Chinese strawberry production, they represented widely grown commercial varieties. This prioritizes cultivars with practical implications for pest management. Detailed information added in section 2.3.
  4. Detailed method about fecundity, pre-adult period and longevity data added in last paragrph of section 2.3.
  5. The second paragraph of section 4.2 rewrite.
  6. Rewriting section 4.3 and 4.4.
  7. Rewriting Conclusion part.

Round 2

Reviewer 1 Report

Comments and Suggestions for Authors

The revised paper is almost ready for publication 

Few typos are met

69: secondatry

71: symbioint, arthropods

93 (elsewhere?): the sentence is started with T.

133: with a condition = at

196-197: no need of plural subject & predicate here

197: Software producer's name or respective reference is expected 

355 (elsewhere?): Latin generic epithets require italics

401: the subject is missing 

Author Response

The revised paper is almost ready for publication Few typos are met Q: 69: secondatry A: Thanks for your comments, the error corrected in manuscript. 71: symbioint, arthropods A: Corrected. 93 (elsewhere?): the sentence is started with T. A: Corrected. 133: with a condition = at A: Corrected. 196-197: no need of plural subject & predicate here A: Corrected. 197: Software producer's name or respective reference is expected A: Detailed information about SPSS 19 added. 355 (elsewhere?): Latin generic epithets require italics A: Corrected. 401: the subject is missing A: Adding “researchers” as subject.

Reviewer 3 Report

Comments and Suggestions for Authors

Upon reviewing the authors’ response and the revised manuscript, I found that the majority of my comments have been adequately addressed, and the manuscript has improved substantially as a result. However, a few of my earlier points remain unaddressed or insufficiently clarified. For example Fig 3, in my opinion, is still confusing, or why authors used non-parametric analyses (Kruskall Wallis) for a particular dataset where it seems a data transformation might have made parametric analysis appropriate. While these are not necessarily major issues, I believe they should be considered before the manuscript can be accepted.

Author Response

Reviewer 3#

Q: Upon reviewing the authors’ response and the revised manuscript, I found that the majority of my comments have been adequately addressed, and the manuscript has improved substantially as a result. However, a few of my earlier points remain unaddressed or insufficiently clarified. For example Fig 3, in my opinion, is still confusing, or why authors used non-parametric analyses (Kruskall Wallis) for a particular dataset where it seems a data transformation might have made parametric analysis appropriate. While these are not necessarily major issues, I believe they should be considered before the manuscript can be accepted.

A: Thank for your comments, and the non-parametric analyses (Kruskall Wallis) was used for data analysis in Fig. 1 and 2. Based on your kind advise, we transformed the data of pre-adult period of Fig. 1 and 2 as follows: y=lg(x). However, the data still do not follow a normal distribution based on Shapiro–Wilk test analyzed by SPSS 19.0. The detailed information in attachment.
